# Immunogenicity of The BNT162b2 COVID-19 mRNA and ChAdOx1 nCoV-19 Vaccines in Patients with Hemoglobinopathies

**DOI:** 10.3390/vaccines10020151

**Published:** 2022-01-20

**Authors:** Osman O. Radhwi, Hamza Jan, Abdullah Waheeb, Sawsan S. Alamri, Hatem M. Alahwal, Iuliana Denetiu, Ashgan Almanzlawey, Adel F. Al-Marzouki, Abdullah T. Almohammadi, Salem M. Bahashwan, Ahmed S. Barefah, Mohamad H. Qari, Adel M. Abuzenadah, Anwar M. Hashem

**Affiliations:** 1Hematology Department, Faculty of Medicine, King Abdulaziz University, Jeddah 21859, Saudi Arabia; hamzajan97@gmail.com (H.J.); abdullahwaheeb@gmail.com (A.W.); halahwal@kau.edu.sa (H.M.A.); afmalmarzouki1@kau.edu.sa (A.F.A.-M.); atalmohammade@kau.edu.sa (A.T.A.); smbahashwan1@kau.edu.sa (S.M.B.); asbarefah@kau.edu.sa (A.S.B.); drqari200@gmail.com (M.H.Q.); 2Hematology Research Unit, King Fahd Medical Research Center, King Abdulaziz University, Jeddah 21859, Saudi Arabia; idenetiu@kau.edu.sa (I.D.); aalmanzlawey@kau.edu.sa (A.A.); 3Vaccines and Immunotherapy Unit, King Fahd Medical Research Center, King Abdulaziz University, Jeddah 21859, Saudi Arabia; salamri0320@stu.kau.edu.sa (S.S.A.); amhashem@kau.edu.sa (A.M.H.); 4Department of Medical Laboratory Sciences, Faculty of Applied Medical Sciences, King Abdulaziz University, Jeddah 21859, Saudi Arabia; aabuzenadah@kau.edu.sa; 5King Fahd Medical Research Center, King Abdulaziz University, Jeddah 21859, Saudi Arabia; 6Department of Medical Microbiology and Parasitology, Faculty of Medicine, King Abdulaziz University, Jeddah 21859, Saudi Arabia

**Keywords:** BNT162b2, ChAdOx1 nCoV-19, COVID-19, vaccine, thalassemia, sickle cell disease

## Abstract

Introduction: Studies assessing immune responses following Pfizer-BioNTech BNT162b2 mRNA COVID-19 (Pfizer) and ChAdOx1 nCoV-19 AZD1222 (AstraZeneca) vaccines in patients with hemoglobinopathy are non-existent in the literature despite being thought at high risk of infection. Methods: Prospectively, we collected serum from patients with hemoglobinopathies at least 14 days post vaccine and measured neutralizing antibodies (nAb) in addition to binding antibodies using in-house assays. Results: All 66 participants mounted a significant binding antibody response (100%), but nAbs were detected in (56/66) post-vaccine with a rate of 84.5%. Age, gender, vaccine type, spleen status, hydroxyurea use, and hyperferritinemia did not affect the rate significantly. While 23/32 (71.8%) patients receiving only one dose of the vaccine were able to mount a positive response, 33/34 (97.05%) of those who had two doses of any vaccine type had a significant nAbs response. Patients who had anti-nucleocapsid (N), signifying asymptomatic infection in the past, were able to produce nAbs (31/31). No nAbs were detected in 10/35 (28.5%) patients with no anti-N antibodies. Conclusion: Our results provide supportive data when advising patients with hemoglobinopathy to receive COVID-19 vaccines and ensure booster doses are available for better immunity. Whenever available, measurement of nAb is recommended.

## 1. Introduction

The severe acute respiratory syndrome coronavirus 2 (SARS-CoV-2) pandemic has affected more than 260 million individuals and resulted in more than 5 million deaths worldwide as of 1 December 2021 [1]. Despite the ongoing global efforts in finding effective therapy against the virus, which is still lacking, protective measures and primary prophylaxis with vaccines remain the most effective strategy. Several novel vaccines have been developed [2]. Four vaccines have been approved for safe use in Saudi Arabia, but only three have been used so far [3]. These vaccines include Oxford–AstraZeneca ChAdOx1 nCoV-19 vaccine (AZD1222) [4] and Pfizer–BioNTech mRNA COVID-19 vaccine (BNT162b2) [5], which gained early access in Saudi Arabia, then followed by the Moderna mRNA SARS-CoV-2 vaccine (mRNA-1273) [6].

Hemoglobinopathies are chronic conditions affecting the production of normal hemoglobin necessary to deliver oxygen and nutrients to various body organs. Patients with hemoglobinopathy, especially thalassemia disorders and sickle cell diseases (SCD), are unique with special clinical care and attention. Cardiac and respiratory conditions are major comorbidities in this population and are the leading cause of death [7]. These patients are at high-risk for COVID-19 complications, with a mortality of 4/162 (2.5%) in SCD and 13/84 (15.5%) in thalassemia patients when hospitalized [8,9,10]. As such, this population of patients should be prioritized for primary prevention of COVID-19 via vaccination as recommended by the centers for disease control and prevention (CDC) [11], the Thalassemia International Federation (TIF) [12], and the National Health Service-England (NHSE) [13].

Currently, little is known about the immunogenicity of COVID-19 vaccines in this population. The lack of knowledge is, in part, related to the exclusion of this population from the clinical trials assessing vaccine efficacy [4,5,6]. Therefore, our aim in this study was to evaluate the immune response to COVID-19 vaccines in patients with hemoglobinopathies by measuring binding and neutralizing antibodies (nAbs).

## 2. Materials and Methods

### 2.1. Clinical Samples

From the period between 27 June and 27 October 2021, we approached patients with hemoglobinopathies presenting to King Abdulaziz University Hospital (KAUH) for their clinical care. Patients older than 18 were included, as this age was allowed to receive the COVID-19 vaccine in Saudi Arabia during this period. Questions around the vaccine dates, vaccine type, use of hydroxyurea, splenectomy, and self-awareness of previous positive COVID-19 PCR tests were collected. Blood samples were collected during the patient’s clinical visit at least 14 days after the vaccination. The group of patients who provided samples in the first visit after the first dose of the vaccine was asked to present again for a second sample fourteen days at least after receiving the second dose of the vaccine. The group of patients who provided samples after the second dose of the vaccine in the first visit was not required to provide additional samples. The enrollment period of the study was four months in total. All participants signed a written informed consent to participate in the study voluntarily. The KAUH ethics committee approved the study (reference number: 332-21).

### 2.2. Laboratory Testing

Serum ferritin levels were assessed using a conventional ELISA kit (Elabscience, Housten, TX, USA, and Abcam, Cambridge, UK) in the hematology laboratory at KAUH. A ferritin level of more than 1000 μg/mL was considered significant. In the vaccines and immunotherapy unit (VIU) at King Fahd Medical Research Centre (KFMRC), end-point titers of COVID-19 binding IgG antibodies against nucleocapsid (N), subunit 1 (S1) of the spike (S) protein, and receptor-binding domain (RBD) were analyzed for post-vaccine samples using in-house ELISA, as previously described [14]. Additionally, nAbs titer was measured using recombinant Vesicular Stomatitis Virus (VSV) pseudovirus expressing SARS-CoV-2 S protein from Wuhan strain (rVSV-ΔG/MERS-S*-luciferase) assay as previously reported [15]. The results of anti-S1 and anti-RBD binding IgG antibodies are reported as end-point titers. The results of anti-S1 and anti-RBD binding IgG antibodies are reported as end-point titers. For nAbs, titers were reported as geometric mean titers (GMT) and were considered positive if GMT is ≥1:40.

### 2.3. Statistical Analysis

Continuous variables were reported as means with ranges, and the two-sample *t*-test was used to compare nAbs producers vs. non-producers. Categorical variables were reported as frequencies, and comparison for significant differences between groups was made using the Fisher’s exact test. Statistical analysis in figures using the Mann–Whitney test. All *p*-values < 0.05 were considered statistically significant. Analyses were performed using Stata version 16.1 (StataCorp) and GraphPad Prism version 9.0.2 software (Graph-Pad Software, Inc., San Diego, CA, USA).

## 3. Results

A total of 83 samples were collected from 66 participants, of which 49 and 34 samples were collected after the 1st or the 2nd dose, respectively. Overall, 32 and 34 patients received one or two doses of any vaccine type, respectively (Table 1).

Among the participants, 51 (77.27%) were diagnosed with transfusion-dependent thalassemia (TDT), and 15 (22.73%) were diagnosed with sickle cell disease (SCD) (Figure 1). Participants’ age ranged from 18–53 years, and 40.9% were males, and 59.1% were females. The median duration from the 1st and 2nd vaccine dose to sample collection were 53 and 61.5 days, respectively (15–136 and 17–155 days). Only 17 individuals provided two samples after the 1st and 2nd doses for an appropriate comparison. In those seventeen individuals, the median duration between vaccination and sample collection was 35.5 (16–136) and 57 (18–111) days for the first and second vaccine doses, respectively. The duration between vaccination and sample collection was not associated with any significant difference in nAbs response for the 1st and 2nd vaccine doses (1st dose: *p* = 0.84, 2nd dose: *p* = 0.18).

Overall, testing for nAbs showed that 56/66 (84.84%) patients mounted a reactive nAbs response after one or two doses. Nonetheless, two doses of any vaccine type resulted in a statistically significant nAb response (33 out of 34 individuals) compared to only one dose of the vaccine (23 out of 32 individuals) (97.05 vs. 71.87%, *p* = 0.005).

Correlated to anti-N IgG, which suggests prior infection, nAbs were evident in 31/31 patients who expressed an anti-N antibody. On the other hand, only 25/35 patients who did not express anti-N IgG were found to have nAbs (100 vs. 71.42%, *p* = 0.001). The levels of nAbs were significantly higher in patients with a previous infection compared to non-infected individuals after a single dose (*p* < 0.0001) or two doses (*p* = 0.0477), in which all participants with previous infection history showed seroconversion (Figure 2a). The second vaccine dose seemed to augment the nAbs response in most individuals despite their previous COVID-19 infection history. Specifically, while 14/25 of the samples collected post-first dose from those who were non-infected were seropositive for nAbs, 19/20 samples from the non-infected group had nAbs after the second dose (Figure 2a). Nonetheless, the levels of nAbs were heterogeneous, and many patients had low titers regardless of their previous COVID-19 infection history or the number of vaccine doses they had received.

Interestingly, nAbs titer differs among participants based on the type of vaccine they have received. As shown in Figure 2b, patients who received homologous prime-boost doses of Pfizer vaccine had ~2.8-fold higher titer than those receiving homologous prime-boost doses of AstraZeneca vaccine. While heterologous prime-boost doses induced higher nAbs response than homologous prime-boost doses of AstraZeneca vaccine, the sequence of the administration of these vaccines seems to affect the outcomes. Specifically, the Pfizer–AstraZeneca prime-boost regimen elicited 3-fold higher nAbs levels than the AstraZeneca–Pfizer prime-boost regimen (Figure 2b).

All participants showed seroconversion and produced anti-S1 and anti-RBD IgG antibodies post-vaccination regardless of their previous exposure to SARS-CoV-2 or the number of doses they received (Figure 2c,d and Appendix A).

Evaluation of the change in nAbs response between the first and second dose of the vaccine showed that only 2 out of the 17 patients (11.76%) who provided their 2nd blood sample could not produce nAbs after the first dose (Figure 3a). These two individuals were from the non-infected group. While all patients had a marked increase in their nAbs titer post 2nd vaccine dose, one patient showed a reduction in his nAbs response post 2nd dose (collected after 36 days) compared to his nAbs response post 1st dose (collected after 133 days) (Figure 3a). Similarly, all 17 participants showed an increase in anti-S1 and anti-RBD IgG binding antibodies post-vaccination from both non-infected and previously infected groups (Figure 3b,c).

Gender, age, type of hemoglobinopathy, spleen status, the use of hydroxyurea, and hyperferritinemia did not affect the nAbs significantly (Table 1). In addition, a sub-group analysis for patients with TDT or SCD did not reveal different findings (Appendix A). An analysis of the relation between vaccine doses and sample collection before or after 3-month duration did not find any statistically significant relation to measured antibody responses (Appendix A).

## 4. Discussion

Our data show nearly 85% of patients with hemoglobinopathies can mount a nAbs response to COVID-19 vaccines. However, this response is markedly affected by the history of previous infection and the number of vaccine doses, where two doses are better than a single dose. Importantly, the response was mostly heterogeneous, with many participants producing low to medium levels of nAbs even after receiving two doses of the vaccine. Surprisingly, anti-N IgG was detected in 46.9% of patients who denied positive COVID-19 testing in the past. This study was conducted after Saudi Arabia went through a SARS-CoV-2 transmission peak with reported high seroprevalence in asymptomatic individuals (30–56.5%) [16,17], which can partially explain this finding.

No previous reports showed the different effects of SARS-CoV-2 variants infection on hemoglobinopathy patients. In Saudi Arabia, only one reported study in the literature provided insight into the prevalence of SARS-CoV-2 variants in the Saudi population. By sequencing samples from 320 patients with COVID-19 infection between April and June 2021, Alhamlan et al. detected the delta variant in 40.9%, beta variant in 15.9%, alpha variant in 11.6%, and the rest were variants of no concerns per authors at that time period [18].

The previous exposure was associated with a better immune response to the vaccine in our study (Figure 2a). These data are consistent with several previous reports in healthy individuals showing that lower nAbs titers or seroconversion rates are seen in naive people compared to those with a history of prior infection [19,20].

The availability of different COVID-19 vaccines eases access to 2nd and even 3rd boosters. In fact, our study shows heterologous vaccination is more immunogenic than homologous vaccination, particularly when Pfizer is followed by AstraZeneca vaccination. In Saudi Arabia, the Public Health Authority allowed vaccine interchangeability under exceptional situations according to the availability of the vaccine or if anaphylactic reactions occurred with the first dose [21], similar to Health Canada [22]. A growing number of small cohort studies have been published looking into the immunogenicity of heterologous vaccinations. An interim analysis of a prospective study assessing the effectiveness of heterologous vaccination in Germany showed AstraZeneca/Pfizer prime-boost to have a higher capacity (*p* < 0.0001) to neutralize alpha and beta variants of COVID-19 virus when compared to Pfizer or AstraZeneca homologous series of vaccination [23]. Moreover, a multicentric UK study looking at the immunogenicity of heterologous vaccination with Pfizer and AstraZeneca versus homologous vaccination showed non-inferiority for Pfizer/AstraZeneca over Pfizer/Pfizer vaccinee with mean anti-spike IgG antibody concentration at 7133 ELU/mL and 14,080 ELU/mL, respectively [24].

Associated clinical aspects around TDT can explain the suboptimal immune response to COVID-19 infection with severe clinical outcomes. This includes, but is not limited to, immunosuppression secondary to spleen removal and iron deposition on tissues causing end-organ damage [25]. Nevertheless, analyzing our study cohort with 46.9% splenectomized patients and 81.8% patients with hyperferritinemia did not find a significant relationship between these two major factors and the ability to mount a reactive nAb response.

On the other hand, hydroxyurea (HU) is an essential medication for patients with SCD; it decreases acute painful events, prevents splenic dysfunction [26] and reduces mortality [27]. It is variably used for TDT patients in our institution, given the controversial benefit [28]. In one of the pivotal randomized controlled trials, HU use in infants and children with sickle cell disease did not affect the immune responses after vaccinations with mumps, measles, rubella, and pneumococcal vaccines [29]. In a more recent paper, the use of HU did not affect the immune responses to yellow fever vaccines in 17 children with sickle cell disease who were on HU compared to 35 children with sickle cell disease who were not on it [30]. Concerning COVID-19 infection, Mucalo et al. reported the outcomes of 750 illness cases of COVID-19 in patients with sickle cell disease. In patients older than 18 years, HU was not shown to affect the incidence of serious COVID-19 illness (CI 0.49–1.51, *p* = 0.59) and hospitalization (CI 0.77–1.10, *p* = 0.35). Moreover, HU showed a trend towards lowering the number of acute pain visits to the emergency department for patients with SCD affected by COVID-19 (CI 0.76–0.98, *p* = 0.02). Similar findings were seen in the pediatric age group [31]. HU is also used for myeloproliferative neoplasms (MPN) for its cytotoxic effect. A recent report by Kozak et al. showed that one out of 5 patients with essential thrombocythemia and one out of 6 patients with polycythemia vera could not mount COVID-19 binding antibodies after two doses of mRNA vaccines [32]. In our cohort, 15 patients (11 SCD and 4 TDT) were on HU and its use did not alter the nAbs response significantly.

Given the possibility of 15.5% not developing nAb, we enforce the importance of complying with strict protective precautionary measures to prevent COVID-19 infection in patients with hemoglobinopathies. Holistically, this should include caution with patients’ frequent visits to healthcare facilities and the use of public transportation [33]. Additionally, a 3rd booster dose is recommended whenever available to ensure persistence of immunity against SARS-CoV-2.

## 5. Limitations

Our study was limited by the small sample size and the lack of a control group to assess the associated risk factors accurately. Additionally, before receiving vaccine doses, measuring anti-N at baseline would have provided additional sights regarding its relationship with the nAbs response.

## 6. Conclusions

Physicians responsible for caring for patients with hemoglobinopathies should advise for serial COVID-19 vaccination and address the possibility of vaccine failure. Whenever available, assessing immune responses by testing for nAb is recommended and not limited to binding antibodies.

## Figures and Tables

**Figure 1 vaccines-10-00151-f001:**
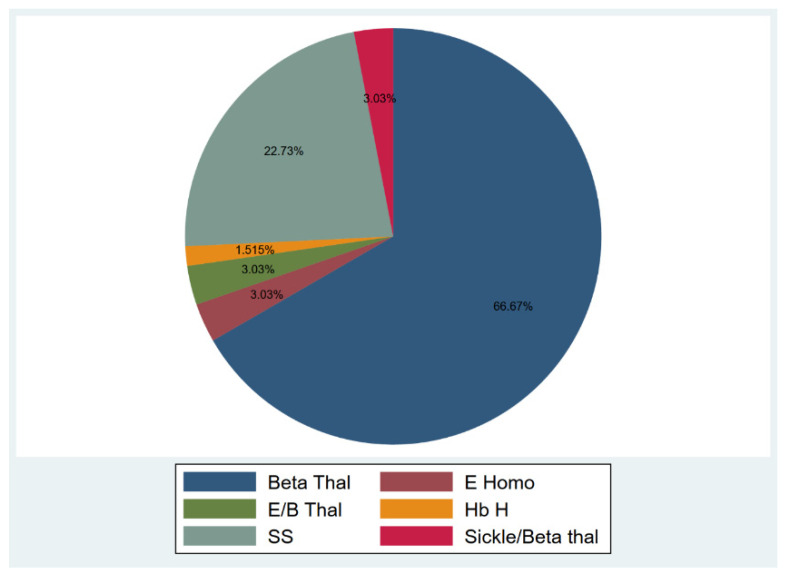
Hemoglobinopathies distribution.

**Figure 2 vaccines-10-00151-f002:**
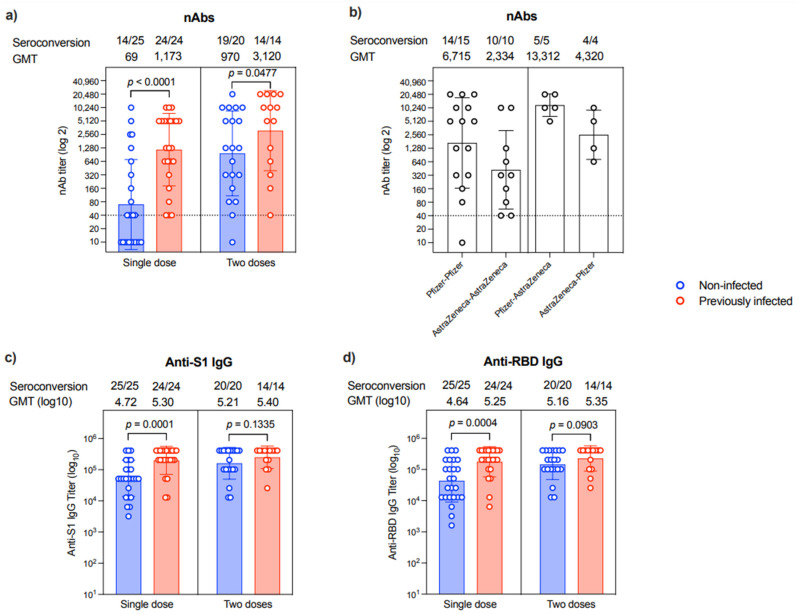
Antibody response. (**a**) Level of nAb response as end-point titer after a single dose or two doses of the different COVID-19 vaccines in non-infected (n = 25 in single-dose group and *n* = 20 in two-dose group) or previously infected (*n* = 24 in single-dose group and *n* = 14 in two-dose group) individuals. (**b**) Levels of nAb response as end-point titer after two doses of the different homologous or heterologous combinations of Pfizer and AstraZeneca COVID-19 vaccines. (**c**) Levels of anti-S1 binding IgG response as end-point titer after a single dose or two doses of the different COVID-19 vaccines in non-infected (*n* = 25 in single-dose group and *n* = 20 in two-dose group) or previously infected (*n* = 24 in single-dose group and *n* = 14 in two-dose group) individuals. (**d**) Levels of anti-RBD binding IgG response as end-point titer after a single dose or two doses of the different COVID-19 vaccines in non-infected (*n* = 25 in single-dose group and n = 20 in two-dose group) or previously infected (*n* = 24 in single-dose group and *n* = 14 in two-dose group) individuals. Data are reported as geometric mean titer ± standard deviation. Numbers of seroconverted individuals and mean titers are shown on the top of the panels. Statistical analysis was done using the Mann–Whitney test. The dotted line in (**a**,**b**) is the cut-off at 1:40.

**Figure 3 vaccines-10-00151-f003:**
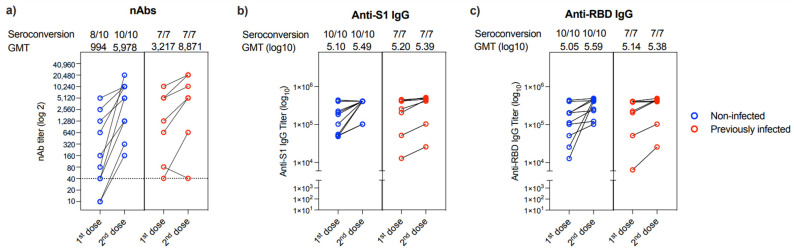
Changes in antibody response in people provided two samples. Changes in (**a**) nAb titers, (**b**) anti-S1 binding IgG end-point titer, and (**c**) anti-RBD binding IgG end-point titer between 1st and 2nd doses in non-infected (*n* = 10) or previously infected (*n* = 7) individuals who provided two samples after vaccination. Data are reported as geometric mean titer ± standard deviation. Numbers of seroconverted individuals and mean titers are shown on the top of the panels. The dotted line in a is the cut-off at 1:40.

**Table 1 vaccines-10-00151-t001:** Patient characteristic.

Characteristic	Category	Total *n* = 66 (%)	Seropositive *n* = 56 (%)	Seronegative *n* = 10 (%)	*p* Value
Gender ^a^	Male	27 (41)	23 (41.1)	4 (40)	1.0
	Female	39 (59)	33 (58.9)	6 (60)	
Age ^b^			31 (18–49)	33.7 (21–53)	0.34
No. of doses ^a^	One	32 (48)	23 (41.1)	9 (90)	0.005
	Two	34 (52)	33 (58.9)	1 (10)	
Hemoglobinopathy ^a^	TDT	51 (77)	43 (76.8)	8 (80)	1.0
	SCD	15 (23)	13 (23.2)	2 (20)	
Splenectomy ^a^	Yes	31 (47)	25 (44.6)	6 (60)	0.5
	No	35 (53)	31 (55.4)	4 (40)	
Hydroxyurea ^a^	Yes	17 (26)	15 (26.8)	2 (20)	1.0
	No	49 (74)	41 (73.2)	8 (80)	
Hyperferritinemia (>1000 μg/mL) ^a^	Yes	54 (82)	45 (80.4)	9 (90)	0.7
	No	12 (18)	11 (19.6)	1 (10)	
History of COVID-19 ^a^; Anti-N positive	Yes	31 (47)	31 (55.4)	0 (0)	0.001
	No	35 (53)	25 (44.6)	10 (100)	

^a^*n* (column percentage); ^b^ Mean (range). TDT = transfusion-dependent thalassemia, SCD = sickle cell disease.

## Data Availability

Raw data are available upon request from the corresponding author.

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
