# Peer review of "Immunogenicity of The BNT162b2 COVID-19 mRNA and ChAdOx1 nCoV-19 Vaccines in Patients with Hemoglobinopathies"

_vaccines, 2022, doi:10.3390/vaccines10020151_

Round 1

Reviewer 1 Report

In this paper a small cohort of patients affected by different hemoglobinopathies (thalassemia, Sickle cell disease) undergoing different therapeutic approaches is studied in term of anti-SARS-CoV-2 antibody production after COVID-19 vaccination. The paper is descriptive, takes into account exclusively determination of serum antibodies after vaccination and the number of patients is limited. However, a favourable point is the description for the first time of the humoral response of vaccinated patients affected by various hemoglobinopathies (albeit limited to the original Wuhan strain) with special emphasis on neutralizing antibodies.

Abstract

  1. ‘No detected nAbs were seen in 10/35 (28.5%) patients who did not have anti-N antibodies’. Please reformulate the sentence.
  2. ‘Our results provide supportive data when advising patients …’. The advise to vaccination does not only reside on reaching immunity but also to the absence of adverse effects. Along with the claim of the Authors, it would be thus questionable the advice to vaccination of patients with humoral immunodeficiencies. Please, reformulate the conclusion.

Introduction

  1. ‘Despite the rapid progress in finding effective therapy against the virus’. I will be more cautious about this affirmation. Although some anti-viral compounds have been recently approved by National Health Services, the real-life demonstration of their effectiveness lacks so far. Second, monoclonal antibodies are of no use in severe respiratory disease which is still orphan of therapies (tocilizumab? Anakinra?).

Methods

  1. ‘Blood samples were collected during the patient's clinical visit at least 14 days after the vaccination’. This sentence and the following one are confusing. Do the Authors mean that blood samples were collected 2 wks after the first dose ? And what does ‘total follow-up period was four months’ means? Could be blood samples also collected after 4 mos after which one vaccination (AZD1222? BNT162b2?).
  2. in-house ELISA was use for the determination of Abs against N,S and RBD proteins referring to a previous paper published in Oct 2020. However, since then several other commercial well-performing assays against the same proteins have been evolved. Thus, a comparison between methods should be included. Cut-off levels for N, S and RBD neither were included in the Section.

Results

  1. ‘Only 17 individuals provided two samples after the 1st and 2nd doses’. Please specify the time intervals between vaccination and sample collection in this really comparative group.
  2. ‘The median duration from the 1st and 2nd vaccine dose to the antibody test…’ 155 days after the second dose could be within the waning period of the antibody response. How do the Authors explain this prolonged interval between vaccination (second dose) and sample collection if samples were claimed as ‘prospectively’ collected (and, thus, necessarily planned for timing) ? Which were the antibody levels (mean or median) in the samples collected 3 mos after the second dose? This point is also of particular concern as patients with hemoglobinopathies might take advantage from a third dose in advance.
  3. The Authors claim that ‘duration between vaccination and sample collection was not associated with any significant difference’ in nAbs measured but they only have 17 individuals who could be strictly (and correctly) compared.
  4. ‘Those who had two doses of any vaccine type had a statistically significant nAb response 33/34 than those who received only one dose of the vaccine 23/32’. Please reformulate this sentence.
  5. ‘one patient showed a reduction in his nAbs response post 2nd dose (Figure 2b).’ Which was the interval between second dose and Abs determination?
  6. In figure 2C it is evident that 5 individuals were heterologously vaccinated with Pfizer+Astra Zeneca. This is unusual and CDC does not recommend this practice (BC Centre for Disease Control. Communicable Disease Control. Vaccine eligibility and registration. Getting your second dose. Vaccine type for second dose for people who received an mRNA vaccine. Updated November 23, 2021.). Please, comment. Further, explanation is due into Discussion section about this point.
  7. Figure 2d and e. Data referred to S and RBD binding Abs in the 17 individuals comparable for the 1st and 2nd dose should be added similarly to Figure 2b or, at least, inserted into the text with values as (data not shown).

Discussion

  1. ‘… after Saudi Arabia went through a SARS-CoV-2 transmission peak with reported high seroprevalence in asymptomatic individuals’. Which SARS-CoV-2 variant was prevalently sequenced in the studied period ?
  2. ‘Mucalo et al. reported HU was not associated with a lower incidence of serious infection and hospitalization’ . Do the Authors mean higher incidence ? The sentence anyhow needs to be corrected.
  3. ‘Additionally, before receiving vaccine doses, measuring anti-N at baseline would have provided additional sights regarding its relationship with the nAbs response’. It is written that

References

  1. Please check that references follow the Author’s guideline of the Journal.

Reviewer 2 Report

As there is no line numbering, I will use the entire paragraph and/or sentence

Introduction:

Comment 1: “These vaccines include Oxford-AstraZeneca ChAdOx1 nCoV-19 vaccine (AZD1222) (4) and Pfizer-BioNTech mRNA COVID-19 vaccine (BNT162b2) (5), which gained early access then followed by Moderna mRNA SARS‐CoV‐2 vaccine (mRNA‐1273) (6)”.

Is this referred to the approved vaccine in Saudi Arabia? Or to the worldwide? If is the first, so rephrase it to be more concise and clear, while if is the second, the references are not appropriate, please use the first call ref.

Comment-2: “Hemoglobinopathies are chronic conditions affecting the production of normal hemoglobin necessary to deliver oxygen and nutrients to various body organs”, is not a complete definition, “ …. Are a group of inherited disorders affecting ….” Should be added.

Ms&Ms

It recommends adding the total number of participants, gender, and the type of vaccine received, as in Saudi Arabia there are three types already in use, per each group of Hemoglobinopathies, even to table1 or S2,3. This will easier for the readers to find out the conclusion correctly.

Comment-3: discussion “The availability of different COVID-19 vaccines eases access to 2nd and even 3rd boosters. In fact, our study shows heterologous vaccination is more immunogenic than homologous vaccination, particularly when Pfizer is followed by AstraZeneca vaccination. An interim analysis of a prospective study assessing the effectiveness of heterologous vaccination in Germany showed AstraZeneca/Pfizer prime-boost to have a higher capacity (20)”.

Without adequate data about the mixed vaccination (the type of vaccine per diseases categories), this paragraph is useless as it seems to belong to “Mix-Match vaccination”.

Recommendation: if the authors have any data concerning the post-COVID-19 vaccination associated symptoms in their vaccinated patient's follow-up should be added to increase the benefits and guidance. whatever,1. a recent comprehensive study considered specific hazard ratios were highest for sickle cell disease patients (7.7-fold) similar for patients with Down's syndrome (12.7-fold increase), kidney transplantation (8.1-fold) (PMID: 34535466). 2. the patients with sickle cell anemia often characterized with high blood viscosity background (PMID: 32656816, PMID: 34838342); and so there is the probable chance for high blood viscosity problem, might be, inducing clot and relative thrombocytopenia (PMID: 34411407)

Round 2

Reviewer 1 Report

The Authors have efficiently and convincingly replied to the major concerns.